# Molecular rearrangement of bicyclic peroxy radicals is a key route to aerosol from aromatics

Siddharth Iyer [1] ✉, Avinash Kumar [1], Anni Savolainen [1], Shawon Barua [1], Christopher Daub [2], Lukas Pichelstorfer[3], Pontus Roldin [4,5], Olga Garmash[1,6], Prasenjit Seal[1], Theo Kurtén [2] & Matti Rissanen [1,2] ✉

The oxidation of aromatics contributes significantly to the formation of atmospheric aerosol. Using toluene as an example, we demonstrate the existence of a molecular rearrangement channel in the oxidation mechanism. Based on both flow reactor experiments and quantum chemical calculations, we show that the bicyclic peroxy radicals (BPRs) formed in OH-initiated aromatic oxidation are much less stable than previously thought, and in the case of the toluene derived ipso-BPRs, lead to aerosol-forming low-volatility products with up to 9 oxygen atoms on sub-second timescales. Similar results are predicted for ipso-BPRs formed from many other aromatic compounds. This reaction class is likely a key route for atmospheric aerosol formation, and including the molecular rearrangement of BPRs may be vital for accurate chemical modeling of the atmosphere.

Aromatic hydrocarbons are an important class of atmospheric trace gases. The global emissions of the most important aromatic molecules, benzene, toluene, ethylbenzene and xylenes (collectively known as BTEX), account for 23% of the global anthropogenic non-methane hydrocarbons on a per-carbon basis[1] and up to 60% of the urban emissions[2,3]. Their sources are primarily anthropogenic, such as from incomplete combustion, industrial processes, and solvent evaporation[4–9]. Recent studies have shown that biogenic sources of BTEX, such as from terrestrial vegetation and from marine phytoplanktons, are also important[10–12], and they have even been measured in the remote Southern ocean and in the arctic[13,14]. Aromatic hydrocarbons in the gas phase pose serious health risks directly[15–18] and also indirectly as their secondary chemistry can lead to tropospheric ozone and secondary organic aerosol (SOA) production[19–23]. The latter can have implications on the planet's radiative balance by affecting the formation and properties of clouds, but significant uncertainties remain regarding the extent of these radiative effects[24–26].

In order to form SOA, participating vapors need to have sufficiently low volatility, which in practice implies molecules with multiple oxygen-containing polar functional groups. Autoxidation[27] has been shown to efficiently form SOA from various aliphatic VOCs, and the molecular level mechanism has been fairly well established for a few key aliphatics[28–31]. Laboratory studies show that aromatics also auto-xidize and form SOA efficiently[22], but all presented molecular level aromatic oxidation mechanisms are far too slow to explain observations. This is a problem because the accurate modeling of SOA from aromatics requires accurately predicting the paths that lead to SOA.

The oxidative chemistry of aromatic hydrocarbons is mainly initiated by OH radicals. In a majority of cases, the reaction of OH with BTEX molecules involves addition to the aromatic ring, with H-atom abstraction playing a minor role at atmospherically relevant temperatures[32]. For toluene, OH addition to the ortho position accounts for 69.8%, followed by ipso, para and meta additions (7.3%, 5.1%, 5.1%, respectively)[33]. The addition reaction leads to the formation of the bridged bicyclic peroxy radical (BPR[34]) as a major product. The

[1]Aerosol Physics Laboratory, Tampere University, FI-33101 Tampere, Finland. [2]Department of Chemistry, University of Helsinki, P.O. Box 55, FI-00014 Helsinki, Finland. [3]Pi-Numerics, 5202 Neumarkt am Wallersee, Austria. [4]Department of Physics, Lund University, P.O. Box 118, SE-221 00 Lund, Sweden. [5]Swedish Environmental Research Institute IVL, SE-211 19 Malmö, Sweden. [6]Department of Atmospheric Sciences, University of Washington, Seattle, WA, USA. ✉e-mail: siddharth.iyer@tuni.fi; matti.rissanen@tuni.fi

critical early steps of toluene oxidation by OH addition are described in Supplementary Section 2. The formation of the BPR from toluene oxidation is considered to be a mechanistic branching point as it has so far been thought to require bimolecular reaction partners to further oxidize due to slow unimolecular reaction rate coefficients (< 0.03 s⁻¹[35]). This is due to the double-ringed structure that sterically hinders both intramolecular H-shift and ring closure reactions forming endoperoxides. In contrast to this mechanistic understanding, flow reactor measurements at 7.9 s reaction time show that the initially formed BPR will quickly add several more oxygen atoms to form peroxy radicals with up to 9 oxygen atoms[35]. These discrepancies suggest an alternative unimolecular fate of the BPR that is both rapid and completely neglected in previous studies.

In this work, we conclusively demonstrate that the toluene derived ipso BPR (i-BPR) is unstable under atmospheric conditions, and undergoes spontaneous molecular rearrangement. This reaction leads to completely ring broken RO₂s, a prerequisite for efficient autoxidation, at rates competitive with bimolecular reactions with NO even under polluted conditions. Intriguingly, several of the organo nitrates and other closed shell products derived from the i-BPR are likewise unstable and decompose on regional atmospheric transport timescales. This could provide, e.g., a transport reservoir for NO$_x$ similar to peroxyacyl nitrates (PAN). Using quantum chemical calculations and master equation simulations that account for energy non-accommodation, we elucidate the molecular level mechanism of the subsequent autoxidation of the ring broken RO₂s that leads rapidly to the highly oxygenated SOA precursor molecules with up to nine oxygen atoms in sub-second timescales. Furthermore, using targeted flow reactor experiments of the OH reaction of toluene and partially deuterated toluene with nitrate chemical ionization mass spectrometry (NO₃-CIMS) detection, we corroborate the proposed mechanism. This is the only unambiguous reaction pathway to toluene derived SOA precursors reported to date.

## Results and discussion

### Spontaneous molecular rearrangement of i-BPR

The rearrangement of i-BPR occurs by the concomitant cleavage of a C-C bond and the O-O bond, along with an H-shift (see Fig. 1a) to lead to a completely ring broken RO₂ with two carbonyl (=O) groups and one hydroxyl (−OH) group. Two rearrangement pathways are possible, C₁ and C₂, depending on which of the two C-C bonds are broken. Relative to the i-BPR, the energy barriers for C₁ and C₂ are 18.8 kcal/mol and 17.9 kcal/mol, respectively, and the energies of the ring broken RO₂s, P-C₁ and P-C₂, are −51.7 kcal/mol and −49.4 kcal/mol, respectively. All energies reported here are zero-point corrected energies. At $P = 1$ atm and $T = 300$ K, the unimolecular rate coefficients of the two pathways are 0.16 s⁻¹ and 0.59 s⁻¹, respectively, which are competitive with RO₂ bimolecular reactions with NO even under heavily polluted conditions. The predicted rate coefficients as a function of temperature are shown in Fig. 1b. Toluene derived BPRs with OH in the ortho, meta and para positions can also undergo the two rearrangement mechanisms, but they are significantly slower and follow the trend of ipso>meta>para>ortho. The rearrangement rate coefficients for meta, para and ortho BPRs are provided in Supplementary Table 7. In addition to being an order-of-magnitude faster than any previously reported isomerization reaction of a toluene derived BPR, the significant exothermicity of the formation of the products means that the rearrangement reactions are irreversible. This is important since some of those previously reported isomerization reactions of BPR, specifically H-shift reactions, were reversible with reverse rate coefficients that are competitive with the bimolecular recombination reactions with O₂ under atmospheric conditions[35].

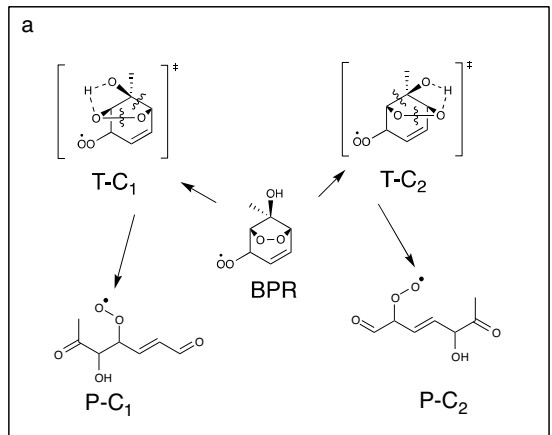

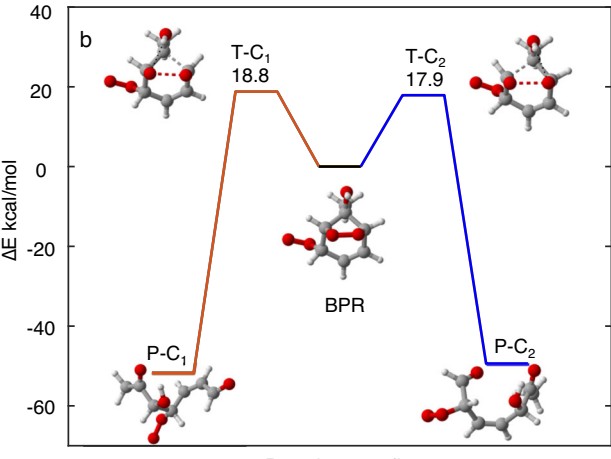

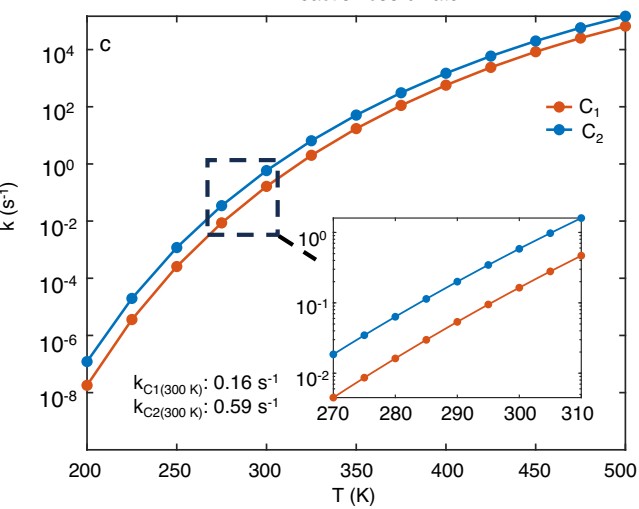

**Fig. 1 | Molecular rearrangement mechanisms of i-BPR. a** 2-D schematic of the two ring breaking mechanisms C₁ and C₂. **b** Stationary points along the potential energy surface of the ring breaking mechanisms with zero-point corrected energies on the y-axis and the reaction coordinate on the x-axis. **c** Master equation derived rate coefficients of the ring breaking mechanisms C₁ and C₂ as a function of temperature (at $P = 1$ atm). Additional points in the atmospherically relevant temperature range are provided in the inset. BPR = bicyclic peroxy radical, T-C₁,₂ = transition states for ring opening pathways 1,2; P-C₁,₂ = corresponding ring broken products. Color coding of atoms: gray - carbon, red oxygen, white - hydrogen.

## Subsequent autoxidation of ring broken peroxy radicals from i-BPR

Both molecular rearrangement mechanisms of i-BPR are expected to occur, leading to the two ring broken $RO_2$s from i-BPR, P-$C_1$ and P-$C_2$. These will subsequently autoxidize rapidly in the absence of the steric hindrance of structural rings, as we have shown previously for the $\alpha$-pinene ozonolysis system[28]. Master equation simulations also show that the excess energy carried by P-$C_1$ and P-$C_2$ allows them to access multiple autoxidation pathways simultaneously. Figure 2 shows the master equation simulation derived initial yields of P-$C_1$ and P-$C_2$, and the yields of the three intermediates formed from the lowest energy barrier autoxidation pathways of P-$C_2$: $R_1$, $R_2$ and $R_3$. The corresponding intermediates from P-$C_1$ were also included in the master equation simulation, but are excluded in Fig. 2. The three autoxidation pathways of P-$C_2$ are shown with differently colored arrows for clarity. Green is the dominant pathway (59%) because the 1,6 H-shift leading to the alkyl radical $R_1$ has the lowest energy barrier. $R_1$ can either react with $O_2$ to lose $HO_2$ and form the carbonyl compound $C_7H_8O_5$, or add an $O_2$ to form the peroxy radical $R_{1b}$-$RO_2$. $R_{1b}$-$RO_2$ will isomerize via a 1,6 H-shift to form the $R_{1b}$-OOH-vinoxy radical, that subsequently adds an $O_2$ to form the $O_9$-$RO_2$ $C_7H_9O_9$. The branching between the carbonyl and peroxy radical pathways is estimated in Supplementary Section 8. The pathway indicated by blue arrows has the next highest yield (16%), and can lead to the $O_{11}$-$RO_2$ $C_7H_9O_{11}$. Finally, the pathway with red arrows has the lowest initial yield, but can subsequently isomerize rapidly to form the $O_7$-$RO_2$ $C_7H_9O_7$ ($R_3$-Epo-$RO_2$), followed by a 1,5 H-shift with a relatively slower rate coefficient to form the $O_9$-$RO_2$ ($R_3$-Epo-OOH-R"$O_2$). $R_3$ and $R_3$-Epo-R' both have small reverse barriers, but the yield of $R_3$-Epo-$RO_2$ is nevertheless likely to be high due to the excess energy of P-$C_2$ as shown in our simulations (see Supplementary Section 12). Note that the last radical intermediates shown for each of the three pathways can continue to autoxidize to form even higher oxygenated products. Due to difficulties in accurately treating the transition states of H-shift reactions from hydroxy groups computationally (see Methods and ref. [31]), the yield of $R_3$ was calculated at a lower level of theory. Therefore, the final yields of $R_1$, $R_2$ and $R_3$ shown in Fig. 2 likely have high uncertainties.

### Experimental results

$NO_3$-CIMS coupled to a flow reactor was used to explore the formation of the oxidized products from the reaction of OH with toluene and the partially deuterated $CD_3$-toluene, where the methyl H-atoms are substituted with D-atoms. Experiments carried out at the short residence times (RT) of 0.8–3.7 s, show that $C_7H_9O_9$ ($O_9$-$RO_2$) with the nominal mass-to-charge ratio of 299 Th is rapidly formed. It is the dominant product peak already at 1.5 s RT, likely due to a combination of two factors: 1) its accumulation at short residence times and 2) the higher sensitivity of the $NO_3$-CIMS for the higher oxygenated products[36]. The recorded spectra and product signals during the 0.8 and 1.5 s RT experiments are shown in Supplementary Figs. 1, 2, while Fig. 3 shows the same during the 3.7 s RT experiment. Even higher oxygenated $C_7H_9O_{11}$ $RO_2$ was also detected at this RT, albeit at much lower signal.

The rapid formation of $O_9$-$RO_2$ agrees with the proposed mechanism from our quantum chemical calculations. Identical results are observed for the partially deuterated $CD_3$-toluene system, which is in line with the proposed mechanism where the methyl group plays no role in the formation of the oxidized products. This is in disagreement with the previous mechanistic understanding where Wang et al. showed that the only unimolecular reaction that could potentially be competitive under atmospheric conditions is the H-abstraction from the methyl group for the toluene derived BPR with OH in the para position[35]. They also reported that the subsequent peroxy radical will quickly lose the abstracted methyl H-atom to OH loss with a rate coefficient of 8 s$^{-1}$ at 298 K, forming a closed-shell carbonyl compound.

However, their computed rate coefficient of 0.026 s$^{-1}$ for the H-abstraction would indicate that bimolecular reactions with NO is likely the dominant sink of the BPR. It is also not sufficiently fast to completely explain their measurement of the 9-oxygen containing $RO_2$ ($O_9$-$RO_2$) from toluene + OH at their 7.9 s experimental residence time. Nevertheless, if this is the dominant fate of the toluene + OH system in our experiments, it would compete with the mechanism proposed in this study. However, our $CD_3$-toluene experiments showed that the dominant closed-shell and peroxy radical products detected contained all three D atoms (see Supplementary Fig. 11), indicating that the mechanism proposed in this work is likely to be the dominant unimolecular pathway to the SOA precursors that we detect. Note that D-shift rates are slower than H-shift rates due to isotope effects, in particular lower tunneling factors, so the use of $CD_3$-toluene will to an extent underestimate the role of H-shift from the methyl group. However, we observe that the mass spectra for toluene and $CD_3$-toluene are near identical, indicating that H-shift from the methyl group is unlikely to play a major role in our experiments. Nevertheless, this pathway might well be important under certain atmospheric conditions. The methyl H-abstraction pathway reported in ref. [35] and the rearrangement pathway reported here do not compete with each other but depend on the site of the initial OH addition to toluene (para vs ipso relative to the methyl substituent). $D_2O$ experiments revealed that the dominant peak corresponding to $O_9$-$RO_2$ contains only two exchangeable H-atoms. This is more consistent with the mechanism we propose because the formed $O_9$-$RO_2$ via the dominant green pathway in Fig. 2 has two groups with exchangeable H-atoms. Following the previously proposed mechanism, $RO_2$ with three exchangeable H-atoms should form, which is not the case. The blue pathway in Fig. 2 does lead to a $O_9$-$RO_2$ with three exchangeable H-atoms, but likely due to its fast isomerization reaction (~230 s$^{-1}$) rapidly forms $O_{11}$-$RO_2$, and is not measured. In addition, the $D_2O$ experiments also indicated two isomers of the $O_7$-$RO_2$, with one and two D-atoms, respectively (see Supplementary Figs 14, 15). While the latter is expected in previous and in our mechanisms, the former can occur if the autoxidation follows the red pathway in Fig. 2 that leads to an $O_7$-$RO_2$ with one exchangeable H-atom. Note that autoxidation is known to produce many isomers and the pathways shown in Fig. 2 are not the only autoxidation pathways occurring in our measurements. However, these are the only pathways that can currently explain the observed H-to-D shifts in the isomers we measure. Details of the $D_2O$ experiments are provided in Supplementary Section 5.

To confirm that these compounds are in fact peroxy radicals, NO was added to form the corresponding organo nitrate ($RONO_2$) products. These experiments clearly illustrated the occurrence of the expected organo nitrates (see Supplementary Figs 12, 13). Reactions of peroxy radicals with NO can also lead to alkoxy radicals and the molecule can continue to oxidize via the alkoxy-peroxy autoxidation pathway. However, the resulting peroxy radicals will have an even number of oxygen atoms and not odd as the peroxy radicals involved in the formation of the organo nitrates reported in this study. The peroxy radicals are unlikely to undergo two NO reactions to revert to an odd number of oxygen atoms followed by a third NO reaction to produce the organo nitrates we detect under the short residence time conditions of our experiments.

The evolution of the oxidation products in the flow tube measurements was also modeled using the Aerosol Dynamics gas- and particle-phase chemistry model for laboratory CHAMber (ADCHAM) studies[37]. This model couples the i-BPR rearrangement pathways described in this work with the conventional Master Chemical Mechanism chemistry of toluene, and achieves a reasonable agreement with the flow reactor data despite the usage of lumped rate coefficients and uncertain branching ratios for the different pathways in toluene oxidation. Details are provided in Supplementary Section 9.

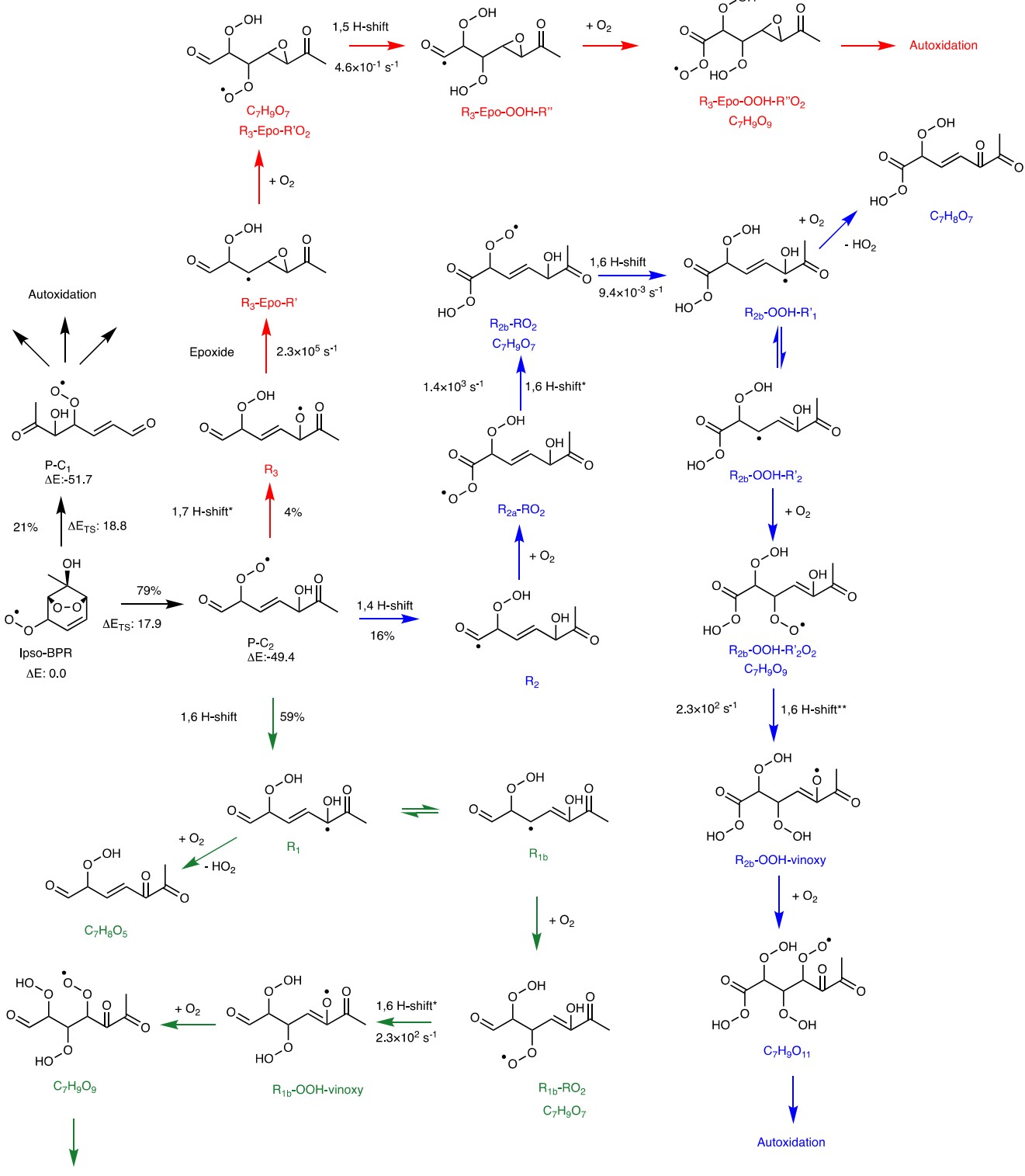

**Fig. 2 | Autoxidation pathways following the molecular rearrangement reactions of i-BPR.** The excess energy from the formation of P-C$_2$ allows it to access multiple autoxidation pathways. Note that P-C$_1$ and P-C$_2$ yields are 21% and 79%, respectively, and the fates of P-C$_2$ sum to 79%. *Rate coefficients calculated at the ωB97X-D/aug-cc-pVTZ level of theory, **Rate coefficient calculated at the B3LYP/6- 31+G(d) level of theory due to the large size and number of conformers for this system. The rest are calculated at the ROHF-ROCCSD(T)-F12a/VDZF12/ωB97X-D/ aug-cc-pVTZ level of theory. Method details with justifications for the different methods employed are provided in Methods.

## Universal rearrangement mechanisms of aromatics
Quantum chemical calculations indicate that similar rearrangement mechanisms are key players in aromatic oxidation in general. Figure 4 shows the rate coefficients of the fastest of the two possible rearrangement mechanisms, C$_1$ and C$_2$ as shown in Fig. 1a, for select key aromatics. Like toluene, the rearrangement mechanism for the other studied aromatic molecules is fastest for i-BPRs, and for molecules, such as meta-xylene and isopropylbenzene, the rate coefficient

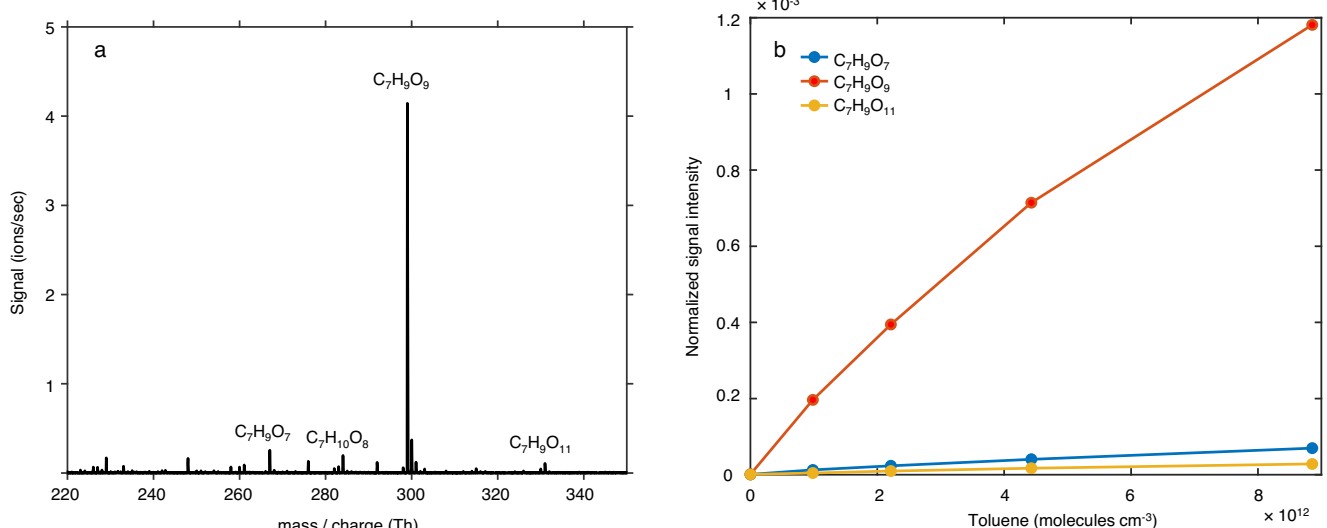

**Fig. 3 | NO₃-CIMS experiments of the reaction of OH radicals with toluene. a** Mass spectra and **b** normalized signals of $RO_2$ radicals as a function of toluene concentration. Reactant concentrations (unit: molecules $cm^{-3}$): [TME] = $1.23 \times 10^{12}$, [O₃] = $5.66 \times 10^{12}$, [Toluene] in Fig. 3a = $8.86 \times 10^{12}$. Reaction time = 3.7 s.

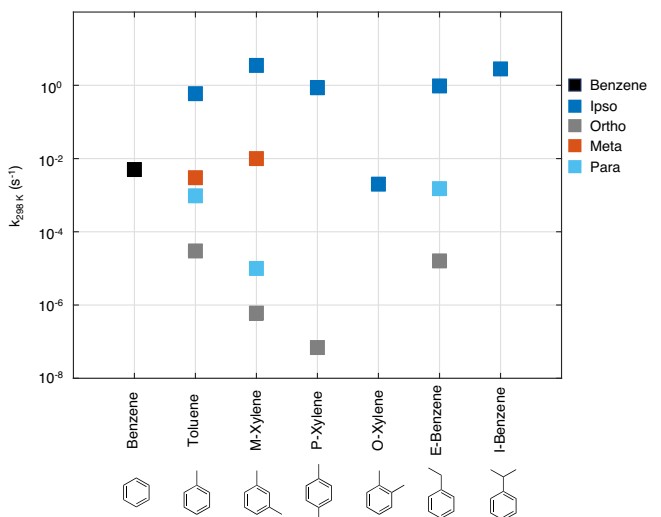

**Fig. 4 | Rate coefficients of the molecular rearrangement mechanisms of select aromatics.** $T$ = 300 K. Molecules studied are benzene, toluene, meta-xylene, para-xylene, ortho-xylene, ethylbenzene and isopropylbenzene. The rate coefficients for BPRs with OH in the meta position for ethylbenzene and in the non-ipso positions for ortho-xylene and isopropylbenzene were not calculated. i-BPRs consistently have the fastest rate coefficients and are competitive under atmospheric conditions for all studied aromatic molecules.

is in excess of $1\,s^{-1}$. For all studied aromatic molecules, the rearrangement mechanism of i-BPRs is their dominant loss process under most atmospheric conditions, and will lead to ring broken $RO_2$s that carry significant excess energy. The subsequent autoxidation of such $RO_2$s will be different from what is currently known, as has been demonstrated here for toluene. Modeling the SOA formation from these aromatic molecules will therefore necessitate the study of their molecular rearrangement mechanisms, and the role of excess energy in their autoxidation. The findings of this study will be a crucial starting point for similar studies on the oxidation of other aromatic molecules, and also for other VOC systems containing structures that could be prone to similar rearrangements (e.g., biogenic terpenes with several cyclic structures and unsaturated sites).

## Atmospheric perspective of the molecular rearrangement mechanism

Field and laboratory measurements show that the oxidation processes of aromatic molecules in general, and toluene in particular, yield SOA. However, the autoxidation mechanism leading unambiguously from toluene + OH to SOA precursors has not been reported thus far. A key challenge for establishing such a mechanism is that bicyclic peroxy radicals from toluene are sterically prevented from taking part in autoxidation. The molecular rearrangement mechanism is the dominant fate of toluene derived i-BPR, and leads directly, and in substantial yield, to hitherto undiscovered peroxy radicals P-C₁ and P-C₂ without steric constraints. These peroxy radicals have access to fast unimolecular reaction channels, and to significant excess energy, allowing them to form peroxy radicals with up to nine oxygen atoms in subsecond timescales and via multiple concurrent autoxidation channels. Our calculations also show that the molecular rearrangement mechanism is operative for bicyclic organo nitrates, and lead to organo nitrates that are completely ring broken (see Supplementary Section 7). In some cases, they can subsequently lose NO₂ and convert back to a radical. This could re-initiate autoxidation and provide a transport mechanism for $NO_x$, similarly as PAN do.

The molecular rearrangement mechanism reported in this work explains the rapid formation of highly oxidized organic products in one of the most important urban SOA-forming systems, toluene + OH, with potential relevance also for other atmospherically abundant aromatic molecules.

## Methods

Systematic conformer sampling of the reactants, intermediates, transition states and products studied here were performed using the MMFF method in Spartan '18 (Wavefunction, Inc.) program. Radical oxygen and carbon atoms in open-shell systems were assigned as uncharged by using the keyword FFHINT On=+-0 for oxygen atoms and FFHINT Cn=+-0, respectively, where n is the atomic label of the relevant O or C atoms. For the non-transition state systems, the initial geometry optimization was carried out using the density functional theory (DFT) method B3LYP/6-31+G(d). For the ring broken systems with many tens to hundreds of possible conformers, a single-point energy calculation was carried out using the B3LYP/6-31+G(d) method, and conformers within 5 kcal/mol in relative electronic energies of the lowest-energy conformers were carried over for the geometry optimization step at the same level of theory. Conformers within 2 kcal/

mol in relative electronic energies after the geometry optimization step at B3LYP/6-31 + G(d) were re-optimized at the higher $\omega$B97X-D/aug-cc-pVTZ level of theory. Frequencies were also calculated using this method. These computations were performed with the ultrafine grid using the Gaussian 16 program[38]. Finally, only a single lowest energy conformer for each reactant, intermediate, and product was taken for the final energy refinement step at the ROHF-ROCCSD(T)-F12a/VDZ-F12 level performed using the Molpro 2021.2 and Molpro 2022.2.2 programs[39].

For the transition states (TS), the geometries were first optimized with the bonds relevant for the TS constrained at approximate distances, followed by unconstrained transition state calculations. Both calculations were carried out at the B3LYP/6-31+G(d) level of theory. Once the TS geometry was found, the structure was taken to Spartan '18 (Wavefunction Inc.). Partial bonds were added to the TS relevant bonds and the distances constrained, and a conformer sampling step was performed using the MMFF method. If the number of conformers numbered in the tens, the single-point energy calculation at the B3LYP/6-31+G(d) was performed, and structures within 5 kcal/mol in relative electronic energies were taken for the optimization step. The constrained and unconstrained TS optimizations were repeated, this time for multiple conformers, at the B3LYP/6-31+G(d) level of theory. Transition state conformers within 2 kcal/mol in relative electronic energies were carried over for TS optimization at the $\omega$B97X-D/aug-cc-pVTZ level of theory. For each studied TS, the energy of the lowest energy conformer was refined at the ROHF-ROCCSD(T)-F12a/VDZ-F12 (abbreviated F12) level of theory.

## Rate calculations using multi-conformer transition state theory (MC-TST)

The rate-coefficients of the autoxidation steps of the ring broken peroxy radicals were calculated using the MC-TST method with Eckart tunneling[40,41]. The MC-TST expression is given by the following:

$$k = \kappa \frac{k_B T}{h} \frac{\sum_i^{\text{all TS conf.}} \exp\left(-\frac{\Delta E_i}{k_B T}\right) Q_{TS,i}}{\sum_j^{\text{all R conf.}} \exp\left(-\frac{\Delta E_j}{k_B T}\right) Q_{R,j}} \exp\left(-\frac{E_{TS} - E_R}{k_B T}\right) \quad (1)$$

Here, $\kappa$ is the quantum-mechanical tunneling coefficient, $T$ is the absolute temperature (=298.15 K), $k_B$ is the Boltzmann's constant and $h$ is the planck's constant. $Q_{R,j}$ and $Q_{TS,i}$ are the partition functions of the reactant and transition state conformers j and i, respectively, and $\Delta E_j$ and $\Delta E_i$ are the zero-point corrected electronic energies of the reactant and transition state conformers relative to the lowest energy conformers, respectively. $E_R$ and $E_{TS}$ are the zero-point corrected electronic energies of the lowest energy reactant and transition state conformers. Partition functions were calculated using DFT at the $\omega$B97X-D/aug-cc-pVTZ level of theory, while the reaction barrier ($E_{TS}-E_R$) includes the F12 correction. Only those conformers that are within 2 kcal/mol of the lowest energy conformer at B3LYP/6-31+G(d) level of theory are included in the summations in the numerator and denominator, which has been reported to be adequate[40].

The Eckart approach was used to calculate the tunneling coefficient $\kappa$.[40,41] Briefly, it is based on the conformers connected to the lowest energy TS conformer by an intrinsic reaction coordinate. Barriers are calculated at the F12 level of theory, while the imaginary frequency of the TS is calculated at the $\omega$B97X-D/aug-cc-pVTZ level of theory. Because of issues with the HF calculations converging to different possible solutions, transition state energies for H atom abstraction from -OH and -OOH groups were calculated at the $\omega$B97X-D/aug-cc-pVTZ level of theory (without F12 corrections). Møller et al.[31] estimate that the uncertainty in the rate coefficients would increase to a factor of 100 without the F12 single-point energy corrections. This is relevant for the rate coefficients of the H-abstractions leading to intermediates $R_{1b}$-OOH-vinoxy, $R_{2b}$-RO$_2$, and $R_3$ in Fig. 2.

## Wavefunction stability

The stabilities of the wavefunctions of the two molecular rearrangement transition states of toluene derived i-BPR were checked at the CCSD(T) stage. This was done to ensure that the lowest lying wavefunction was found. First, a CASSCF calculation was carried out with the MINAO basis set, and with the 1s and 2s orbitals of the heavy (non-H) atoms frozen. Included in the active space were, all occupied p-orbitals/electrons of H atoms, the one half-occupied orbitals for radicals, and one virtual (unoccupied) orbital to allow for rotations within the active space. This is then used as input for the RHF calculation with the larger desired basis set. These calculations were carried out with the Molpro 2022.2.2 program[39]. This is a much more robust approach for checking wavefunction stability than e.g., the standard Stable = Opt check in Gaussian. An example input file is provided in the supplementary files.

## Master equation simulations

The master equation solver for multi-energy well reactions (MESMER) program[42] was used to carry out the master equation simulations. The SimpleRRKM method with Eckart tunneling was used to calculate the rate coefficients of the rearrangement reactions, $C_1$ and $C_2$, of toluene and the other aromatics derived i-BPRs. In a separate MESMER simulation, the first isomerization intermediates of the ring broken intermediates P-C$_1$ and P-C$_2$ were included to calculate the final yields. Lennard-Jones parameters for the toluene derived species: $\sigma = 5.92$ Å and $\epsilon = 410$ K, which are the values for toluene[43,44]. The exponential energy decay with an average energy transfer per collision, $\Delta E_{down}$, was given a value of 278.89 cm$^{-1}$, which is based on experimental results for toluene with N$_2$ bath gas[43]. Grain size = 100 cm$^{-1}$ and numerical precision = dd. In the simulations where the yields of the different pathways were calculated, temperature and pressure were given values of 298.15 and 760 Torr, respectively. The temperature was also varied to derive the temperature dependent rates of the two rearrangement reactions $C_1$ and $C_2$ of toluene derived i-BPR.

In the MESMER simulations, the internal modes corresponding to internal rotations were removed from the vibrational frequencies and treated as hindered rotors. This was performed by first running computations using Gaussian 16 to identify the hindered rotors, and to calculate the corresponding periodicities and barriers using the Freq = HinderedRotor keyword at the $\omega$97X-D/aug-cc-pVTZ level of theory. For the subsequent MESMER simulation, the HinderedRotorQM1D method was used, which requires the Gaussian derived periodicities and barriers, the bonds corresponding to the hindered rotors and the removal of corresponding harmonic frequencies.

The MESMER input file used for calculating the yields of the different pathways following the rearrangement of toluene derived i-BPR is provided in the supplementary files.

## Mass spectrometry experiments

Toluene and CD$_3$-toluene were mixed with OH generated by reacting ozone with tetramethylethylene in a quartz flow reactor. The products were subsequently detected by a nitrate-based chemical ionization mass spectrometer (CIMS). The reaction time was controlled by directing the toluene and CD$_3$-toluene flow through an injector tube within the flow reactor, limiting their residence time with OH from 3.7 s to 1.5 s and 0.8 s. For a given OH concentration, the concentrations of toluene and CD$_3$-toluene were varied to observe the changes in product signals. For the NO experiments, toluene/CD$_3$-toluene and OH concentrations were kept constant, while the NO concentration was varied. For the D$_2$O experiments, N$_2$ was bubbled through a D$_2$O reservoir to introduce it into the flow reactor, where it was allowed to mix with the reactants and the oxidation products. The experiments were conducted under atmospheric conditions.

## Data availability

Supplemental information is available for this paper. Quantum chemical output files and plotting scripts are available online through the public research data archive Zenodo (https://doi.org/10.5281/zenodo.8214481). Source data are provided with this paper.

## Code availability

The scripts used to produce the plots are provided under https://doi.org/10.5281/zenodo.8214481.

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

## Acknowledgements

We thank the TofTools team for the data analysis tools and the CSC IT Center for Science in Espoo, Finland, for providing the computing resources. We also thank Nino Runeberg of CSC for help with wave-function stability checks. This project has received funding from the European Reasearch Council under the European Union's Horizon 2020 research and innovation program under Grant No. 101002728 (M.R.). The support from the Academy of Finland 331207, 336531, 346373 (M.R.), the Swedish Research Council VR 2019-05006 (P.R.) and the Swedish Research Council FORMAS 2018-01745 (P.R.) are also acknowledged.

## Author contributions

S.I., A.S., and C.D. performed the quantum chemical calculations. A.K., S.B., S.I., O.G., and M.R. planned and conducted the flow reactor experiments. S.I. and A.K. analyzed the experimental data. L.P. and P.R. carried out the modeling of the flow reactor experiments. S.I. wrote the manuscript. S.I., A.K., S.B., P.R., O.G., P.S., T.K., and M.R. commented on and edited the manuscript.

## Competing interests

The authors declare no competing interests.
