## [Peer Review File · Nature Communications]

Molecular rearrangement of bicyclic peroxy radicals is a key route to aerosol from aromaticsReviewers' Comments:

Reviewer #1:

Remarks to the Author:

This manuscript presents quantum chemical, statistical rate theory, and flow reactor-mass spectrometric results to establish a new mechanism for the generation of secondary organic aerosol (SOA) precursors in the OH-initiated oxidation of aromatics. The novel reaction being proposed is a concerted rearrangement of bicyclic peroxy radicals in which C-C homolysis, O-O homolysis, and transfer of the H atom from the OH to one of the incipient alkoxy O atoms all happen at one transition state. The manuscript is technically solid, the major results are presented clearly, and the discoveries are highly significant for the area of atmospheric chemistry. I recommend its publication after a few minor issues are addressed:

* The labels within Figure 2 are a little hard to read. Please print this figure as large as possible.

* In the caption to Figure 2, the reference to "P-S₂" should probably be to P-C₂.

* One of the O₂ addition barriers reported in Supplementary Section 8 is negative. The same is true in Supplementary Section 2. This does not impede the MESMER simulations based on the quantum chemical data, nor does it render the predicted O₂ addition branching ratios invalid. Still, the authors should acknowledge that this is a technical problem that adds uncertainty to the predicted branching ratios.

* On p. 10 and in Figure 4, there is a consideration of "the fastest of the two possible rearrangements for select key aromatics." I am pretty sure that the authors are referring to the rearrangements depicted in Figure 1, but the authors should be explicit about this.

* The ends of the two paragraphs in the **Atmospheric perspective of the molecular rearrangement mechanism** section are somewhat redundant.

Reviewer #2:

Remarks to the Author:

On NCOMMS-23-12910-T, "Molecular rearrangement of bicyclic peroxy radicals: key route to aerosol from aromatics", by Iyer et al.

This manuscript proposed a new rearrangement of ipso-BPRs and also the bicyclic hydroperoxides and nitrates in oxidation of aromatics. The study suggested a new route for formation of highly oxygenated molecules (HOMs) with supports from quantum chemical and kinetic calculations and mass spectrometric observations. Deuterated toluene and D₂O were used to probe the reaction pathways and to check the exchangeable protons in the products. Calculations were done at adequate levels, and the experiments were designed carefully to sort support for the proposed mechanism.

The finding here might be a significant improvement on our understanding in mechanism of HOMs and thereafter the mechanism of SOA formation.

Comments:

Line 119 "potentially reversible": The reversible H-shifts here should be slow as well, and is much slower than the bimolecular recombination with O₂. Therefore should "also be virtually irreversible".

On reaction of C₆H₅CD₃: There is an issue of isotope effect and tunneling correction here when changing from -CH₃ to -CD₃. Obviously, the H-shifts here should be much faster than the D-shifts. This might be part of the reason for the low signal for C₇D₂H₆O₆ in Fig S11, and probably for C₇D₂H₆O₈ as well. Using -CD₃ might underestimate the role of H shifts in Ref 35 (Wang et al., EST, 51, 8442, 2017).

Fig S16, RONO₂: BPR + NO forms ROONO first, which might decompose to RO + ONO or isomerize to RONO₂. Even though barrier for rearrangement of RONO₂ is lower than RO + NO₂, the scheme here seems to exaggerate the role of rearrangement. RONO₂s were missing in Fig S16 and Table S9.

D₂O Experiments:

(1) Presumably, C₇H₉O₉ and C₇D₂H₇O₉ were from reaction of C₆H₅CH₃, and C₇D₃H₆O₉ and C₇D₅H₄O₉ were from reaction of C₆H₅CD₃ (Table S6). So the C₇H₉O₉ contains only two exchangeable H-atoms, instead "two, three and four" (Page S14). Please confirm or explain.

(2) In Fig 2, three C₉H₉O₉ isomers were given as R₃-Epo-OOH-R'O₂ (containing one exchangeable H), R₂b-OOH-R'O₂ (containing three exchangeable Hs), and the green C₇H₉O₉ (lower left corner, containing two exchangeable Hs). But R₂b-OOH-R'O₂ has a fast isomerization rate of ~230 s⁻¹, and the last isomer would have a fast H shift (from the aldehyde H). Alternative isomers of C₇H₉O₉ might be needed to account for the observed signal.

(3) Page S14, "... Fig S11 shows the time series of the normalized signals ...": The figure cannot be found.

Kinetics of R₁ (as in Fig S17 and S18): It should be noticed that the two additions of O₂ are both reversible, particularly for R₁a-RO₂ and when R₁ contains excess energy. The usual "+ O₂ - HO₂" process in alpha-hydroxyl alkyl radicals may be limited by the 1,4 H-shift because the addition of O₂ to R₁a-RO₂ is much less exothermic (only ~7 kcal/mol here) than the addition to normal alkyl radicals such as CH₃CHOH radical (exothermic by ~30 kcal/mol).

Flow tube simulations: Could typical concentrations of OH radical and RO₂ be given here? These quantities are essential in assessing the competition between unimolecular and bimolecular reactions for peroxy radicals.

Minors:

Page S9, the 4th line from Bottom: "Fig S4" should be "Fig S9"; Page S10, "Fig S5" should be "Fig S10"; Page S11: "Fig S6" should be "S11"; ...

Reviewer #3:

Remarks to the Author:

This manuscript describes a pathway in the atmospheric oxidation of aromatics that appears to be the missing link between the traditional chemistry and the unexplained HOM formation from these compounds. By combining theoretical kinetic calculations, a kinetic model, and experimental observations, the authors make a case that inclusion of this pathway leads to predictions that are consistent with the observations. The research is timely, as there is a lot of interest in HOM and aerosol formation, and the role of aromatics in urban air quality. The methodologies used are of high level and appropriate for the topic, leading to robust results. This paper describes a high-impact result that has the potential to solve a long-standing conundrum. I support publication of this manuscript.

Reviewing the experimental section is outside my field of expertise.

Most of my comments are minor, but I do have 2 main comments that I urge the authors to consider.

Main comments

To me, it feels that the authors did not do due diligence to the literature. While I recognize that space and number of references are limited in the main text, the supporting information contains extensive descriptions of mechanistic steps without any referencing at all. Most of this even seems to be based on a single paper (Wang and coworkers), which does not acknowledge the vast effort that the community has put into elucidating aromatics oxidation mechanisms to establish a base upon which the current study can add its contribution. The paper also generalizes to aromatics beyond toluene, which is certainly not covered by a single reference to Wang et al.

Reviewing the aromatics literature is daunting and I understand that this is not the manuscript to do this in full, but at the very least some key papers should be cited, as well as mechanistic reviews (e.g. Vereecken 2019, Vereecken and Francisco 2012, Atkinson 2003, Calvert 2002,...). These, and perhaps some additional references to SARs, will also support some of the calculations in the SI that were only done using low-level methodologies. Below, I will not list all places where literature citations are needed, as it is pervasive.

One should be weary about presenting the experiments as proof of the contribution of the characterized pathway. The experiments show that an autoxidation chemistry exists, but does not establish that the molecules observed are those predicted by the theoretical analysis; autoxidation is known to generate many isomers. While the results seem consistent, there remain uncertainties on the theoretical predictions and the experimental observations, and the latter only span a small range of reaction conditions that may be insufficient to discriminate between distinct reaction mechanisms. Likewise, the authors do not establish that the proposed mechanism is also consistent with the observations available in the literature.

Related to this is that, at first glance, the mechanism seems to miss some pathways, or at least does not discuss their contribution as far as I could find. This includes e.g. the reversibility of the epoxidation in R3 (competitive against O₂ addition), the H-migration from -OOH to -O-radical in R3 (with a literature rate $\sim 10^{10} \text{ s}^{-1}$), or H-scrambling and ring closure in R2b-OOH-R'2O₂, all of which represent reaction classes known to be fast. The kinetic calculations also do not seem to account for the impact of fast H-scrambling across OOH/OO groups on the rate coefficient, as well-documented in the literature. It is unclear whether any of this would affect the predictions, but perhaps the mechanism is not fully robust yet.

Minor comments

p. 3 line 79: "finite timescales"
define "finite", in this context probably relative to atmospheric transport time scales (regional, continental, hemisphere, global ?)

p. 4 lines 90-105:
This would be significantly easier to understand with a graphical inset with a Lewis-structure representation of an example.

p. 5 fig 1:
Top: One can not really see what the rearrangement is, due to the use of ball-and-stick graphics. See also remark above.
Bottom: Perhaps an Arrhenius plot with x-axis in $1000\text{K}/T$? Then again, Nat.Comm is not a kinetics journal.

p 7 fig 2:
- Caption: "autoxidation" -> autoxidation
- The intermediates must not be represented as trans-alkenes but strictly as cis-isomers or unspecified (where allylic rotation is possible). Trans-stereoisomers are not what was calculated (sampling a few of the log files shows as much), are not what is expected for the parent molecule

decomposition, and would prevent some of the chemistry to happen (e.g. the H-shift in R2b-RO2)
- The fates for P-C2 do not sum to 100%

p. 9 line 183: " indicating that the mechanism proposed in this work is the dominant unimolecular pathway to the SOA precursors that we detect."
"is" -> is likely to be / is consistent with / ...

p. 12, fig 4 : perhaps remove the minor grid lines to lighten the plot

p. 13, line 246: "This could re-initiate autoxidation and provide a long range transport mechanism for NOx"

It is unclear how prompt decomposition losing NO2 could be useful for long-range NOx transport, as the NO2 is then lost before it can be transported?

p. 16, wave function stability

What do the CASSCF calculations show for multi-reference effects on the biradical O-O bond breaking?
Is the active space (1 virtual orbital) large enough to accommodate such multi-reference effects ?

p. 17, line 341 "In the MESMER simulations, the harmonic frequencies were treated as hindered rotors"

I assume the authors mean that only those internal modes corresponding to internal rotations were removed from the vibrational frequencies and treated as hindered rotors.

p S9, top, ring opening in B-alkyl

There are much better discussions of this channel available that should be cited, e.g. Glowacki et al. 2009, Vereecken 2018, Vereecken 2019, as well as experimental evidence of its lack of importance by the Wennberg group (Xu et al. 2020)

p. S12, NO experiments.

NO can lead to autoxidation proceeding through different pathways (so-called alkoxy-peroxy autoxidation steps). They may form different isomers of the same mass, and a mass spectrum won't reveal mechanistic changes. Alkoxy-peroxy autoxidation is also implicated as a key step in ring breaking HOM formation in some VOC (e.g. Shen et al. 2022, Guo et al. 2022). Furthermore, Vereecken (2018, 2019) points out the role of NO in changing the early stage aromatic oxidation mechanism, based on available literature data. These experiments may thus not as straightforward to interpret as the authors imply.

p. S17, top: "the rearrangement mechanism is also possible for other aromatic derived bicyclic molecules. Table S8 "

Without context, it is not clear which rearrangement mechanism is meant. The bicyclic alkyl radicals have very different rearrangement mechanism reported (ring retaining with epoxide-alkoxy) than given in this work for bicyclic peroxy radicals (ring opening with H-shift and carbonyl formation).

p. S19, bottom

- Mention "site-specific O2 addition" at the start of the last paragraph
- "H-abstraction by O2 to form C7H8O5 is not direct but follows the initial addition of O2 to the C(OH) carbon."

This needs specific literature citations

- "These transition states were calculated to be -0.7 kcal/mol and 1.9 kcal/mol, respectively, above R1 at F12 level of theory." -> "...relative to R1 at the F12 level of theory."

- " the potential energy surface shown in Fig. S12 was used as input in a MESMER simulation to account for the excess energy of R1."

This glosses over a lot of details. At the very least refer explicitly to the Mesmer input files in the Zenodo archive, but preferable document this more thoroughly.

p. S20, bottom " the analytical solution is represented by its Taylor expansion"
The analytical solution to what? The MCM ?

p. S21, middle "The likely reason for this is the less reactive TME RO2s."
Why are they less reactive ? Is literature data available ?

p. S22, ref 1: remove "Vertiotie.4P"

References (Alph.)

Atkinson, R. and Arey, J.: Atmospheric Degradation of Volatile Organic Compounds, *Chem. Rev.*, 103, 4605–4638, <https://doi.org/10.1021/cr0206420>, 2003.

Calvert, J. G., Atkinson, R., Becker, K. H., Kamens, R. M., Seinfeld, J. H., Wallington, T. J., and Yarwood, G.: The mechanisms of atmospheric oxidation of aromatic hydrocarbons, Oxford University Press, Oxford, New York, 566 pp., 2002.

Glowacki, D. R., Wang, L., and Pilling, M. J.: Evidence of Formation of Bicyclic Species in the Early Stages of Atmospheric Benzene Oxidation, *J. Phys. Chem. A*, 113, 5385–5396, <https://doi.org/10.1021/jp9001466>, 2009.

Guo, Y., Shen, H., Pullinen, I., Luo, H., Kang, S., Vereecken, L., Fuchs, H., Hallquist, M., Acir, I.-H., Tillmann, R., Rohrer, F., Wildt, J., Kiendler-Scharr, A., Wahner, A., Zhao, D., and Mentel, T. F.: Identification of highly oxygenated organic molecules and their role in aerosol formation in the reaction of limonene with nitrate radical, *Atmos. Chem. Phys.*, 22, 11323–11346, <https://doi.org/10.5194/acp-22-11323-2022>, 2022.

Shen, H., Vereecken, L., Kang, S., Pullinen, I., Fuchs, H., Zhao, D., and Mentel, T. F.: Unexpected significance of a minor reaction pathway in daytime formation of biogenic highly oxygenated organic compounds, *Sci. Adv.*, 8, eabp8702, <https://doi.org/10.1126/sciadv.abp8702>, 2022.

Vereecken, L.: Reaction mechanisms for the atmospheric oxidation of monocyclic aromatic compounds, in: *Advances in Atmospheric Chemistry: Volume 2: Organic Oxidation and Multiphase Chemistry*, edited by: Barker, J. R., Steiner, A. L., and Wallington, T. J., World Scientific Publishing Co. Pte. Ltd., Singapore, 377–527, https://doi.org/10.1142/9789813271838_0006, 2019.

Vereecken, L.: The mechanism of aromatic oxidation: thoughts based on theoretical and experimental data, with alternative reduced mechanism, *Gases/Atmospheric Modelling/Troposphere/Chemistry (chemical composition and reactions)/acp-2018-146*, <https://doi.org/10.5194/acp-2018-146-SC1>, 2018.

Vereecken, L. and Francisco, J. S.: Theoretical studies of atmospheric reaction mechanisms in the troposphere, *Chem. Soc. Rev.*, 41, 6259–6293, <https://doi.org/10.1039/c2cs35070j>, 2012.

Xu, L., Møller, K. H., Crouse, J. D., Kjaergaard, H. G., and Wennberg, P. O.: New Insights into the Radical Chemistry and Product Distribution in the OH-Initiated Oxidation of Benzene, *Environ. Sci. Technol.*, 54, 13467–13477, <https://doi.org/10.1021/acs.est.0c04780>, 2020.

We thank all reviewers for their valuable input that helped improve the manuscript.

Reviewer 1:

General comment: This manuscript presents quantum chemical, statistical rate theory, and flow reactor-mass spectrometric results to establish a new mechanism for the generation of secondary organic aerosol (SOA) precursors in the OH-initiated oxidation of aromatics. The novel reaction being proposed is a concerted rearrangement of bicyclic peroxy radicals in which C-C homolysis, O-O homolysis, and transfer of the H atom from the OH to one of the incipient alkoxy O atoms all happen at one transition state. The manuscript is technically solid, the major results are presented clearly, and the discoveries are highly significant for the area of atmospheric chemistry. I recommend its publication after a few minor issues are addressed:

Author comment: We thank the reviewer for their comments and their recommendation for publication of our manuscript.

1. The labels within Figure 2 are a little hard to read. Please print this figure as large as possible.

Author comment: We have now increased the font size of the labels in Figure 2.

Changes to manuscript: A larger font size is used for the labels in Fig. 2 of the main manuscript.

2. In the caption to Figure 2, the reference to "P-S₂" should probably be to P-C₂.

Author comment: Thank you for pointing this out. This has now been corrected.

Changes to manuscript: "P-S₂" in Figure 2 caption corrected to "P-C₂".

3. One of the O₂ addition barriers reported in Supplementary Section 8 is negative. The same is true in Supplementary Section 2. This does not impede the MESMER simulations based on the quantum chemical data, nor does it render the predicted O₂ addition branching ratios invalid. Still, the authors should acknowledge that this is a technical problem that adds uncertainty to the predicted branching ratios.

Author comment: We agree. This is now acknowledged in Supplementary Section 8. Regarding Supplementary Section 2, we suspect that the reviewer is indicating this line in page S6: "This corroborates with the energy barriers for the syn and anti O₂ additions we calculated at the F12 level of theory - 2.5 kcal/mol and 3.4 kcal/mol, respectively." If so, both barriers are in fact positive (+2.5 kcal/mol and +3.4 kcal/mol), and our use of the hyphen is injudicious here. The hyphen (-) has now been changed to a colon (:). for clarity.

Changes to supplementary:

Supplementary Section 8: "Note that the F12 correction causes TS_i to be slightly negative, and while this does not impede the MESMER simulations, it does add some uncertainty to the predicted branching ratios". Supplementary section 2, page S6: "- " => ":".

4. On p. 10 and in Figure 4, there is a consideration of "the fastest of the two possible rearrangements for select key aromatics." I am pretty sure that the authors are referring to the rearrangements depicted in Figure 1, but the authors should be explicit about this.

Author comment: We agree that this should be more clearly acknowledged. The text on p. 10 of the main manuscript has been modified to make this clear.

Changes to manuscript: Sentence in p. 10 of the main manuscript changed to: "Fig. 4 shows the rate coefficients of the fastest of the two possible rearrangement mechanisms, C_1 and C_2 as shown in Fig. 1 A, for select key aromatics."

5. The ends of the two paragraphs in the **Atmospheric perspective of the molecular rearrangement mechanism** section are somewhat redundant.

Author comment: Thank you for pointing this out, and we agree. We have decided to combine the two paragraphs and modify the last summary sentence.

Changes to manuscript: The two paragraphs in the "**Atmospheric perspective of the molecular rearrangement mechanism**" section have been combined and this modified final summary sentence has been added: "The novel molecular rearrangement mechanism reported in this work explains the rapid formation of highly oxidized organic products in one of the most important urban SOA-forming systems, toluene + OH, with potential relevance also for other atmospherically abundant aromatic molecules."

Reviewer 2:

General comment: This manuscript proposed a new rearrangement of ipso-BPRs and also the bicyclic hydroperoxides and nitrates in oxidation of aromatics. The study suggested a new route for formation of highly oxygenated molecules (HOMs) with supports from quantum chemical and kinetic calculations and mass spectrometric observations. Deuterated toluene and D₂O were used to probe the reaction pathways and to check the exchangeable protons in the products. Calculations were done at adequate levels, and the experiments were designed carefully to sort support for the proposed mechanism.

The finding here might be a significant improvement on our understanding in mechanism of HOMs and thereafter the mechanism of SOA formation.

Author comment: We thank the reviewer for their comments.

1. # Line 119 "potentially reversible": The reversible H-shifts here should be slow as well, and is much slower than the bimolecular recombination with O₂. Therefore should "also be virtually irreversible".

Author comment: In their 2017 paper, Wang et al. (Wang et al. *Environ. Sci. Technol.* 2017, 51, 15, 8442–8449) report in Table 1 reverse rates for H-migrations in toluene derived BPRs of $1.9 \times 10^6 \text{ s}^{-1}$ and $2.8 \times 10^8 \text{ s}^{-1}$, which are competitive with bimolecular recombination reactions with O_2 under atmospheric conditions. These are for H-migration reactions of BPRs formed from OH additions to ipso and ortho positions, however, and they report virtually irreversible H-migration reactions for the BPR with OH in the para position. We have now modified our text to say that only *some* of the reported isomerization reactions of BPR are irreversible.

Changes to manuscript: Sentence in the main manuscript modified to: *“This is important since some of those previously reported isomerization reactions of BPR, specifically H-shift reactions, were reversible with reverse rate coefficients that are competitive with the bimolecular recombination reactions with O_2 under atmospheric conditions.”*

2. # On reaction of $\text{C}_6\text{H}_5\text{CD}_3$: There is an issue of isotope effect and tunneling correction here when changing from $-\text{CH}_3$ to $-\text{CD}_3$. Obviously, the H-shifts here should be much faster than the D-shifts. This might be part of the reason for the low signal for $\text{C}_7\text{D}_2\text{H}_6\text{O}_6$ in Fig S11, and probably for $\text{C}_7\text{D}_2\text{H}_6\text{O}_8$ as well. Using $-\text{CD}_3$ might underestimate the role of H shifts in Ref 35 (Wang et al., *EST*, 51, 8442, 2017).

Author comment: We agree that the slower $-\text{CD}_3$ shift rate could be part of the reason for the observed low $\text{C}_7\text{D}_2\text{H}_6\text{O}_6$ signals. If the methyl H-shift was a dominant reaction in our $\text{C}_6\text{H}_5\text{CH}_3$ measurements and missing in our $\text{C}_6\text{H}_5\text{CD}_3$ measurements, the observed mass spectra for the two precursors would be expected to be different. However, we observe that the produced mass spectra are near identical (Figs. S12 to S15). We now include the signals of the closed-shell products from the $\text{C}_6\text{H}_5\text{CH}_3$ experiment (Fig. S11 B) in the supplementary. These signals are also low, indicating that the methyl H-shift is unlikely to play a major role under our experimental conditions. We agree that the isotope effect should nevertheless be noted. We now note it in the manuscript and supplementary text.

Changes to manuscript: We have noted in the manuscript that the use of CD_3 -toluene might underestimate the role of methyl H-shifts. *“Note that D-shift rates are slower than H-shift rates due to isotope effects, in particular lower tunneling factors, so the use of CD_3 -toluene will to an extent underestimate the role of H-shift from the methyl group. We observe that the mass spectra for toluene and CD_3 -toluene are near identical, indicating that H-shift from the methyl group is unlikely to play a major role.”*

Changes to supplementary: *“While D-shift reactions are known to be slower than H-shifts, and the CD_3 -toluene experiments could underestimate the role of methyl H-shifts, we also measured low signals of the closed-shell species from the CH_3 -toluene experiments (Fig. S11 B), indicating that the methyl H-shift is unlikely to play a major role under our experimental conditions.”*

Fig. S11 caption: Fig. S11 B added that shows the signals of the closed-shell species from the CH_3 -toluene experiment. Text added to Fig. S11 caption: *“The closed-shell species are also low in the CH_3 -toluene experiment, indicating that H-shift from the methyl carbon is playing an insignificant role under our experimental conditions.”*

3. Fig S16, RONO₂: BPR + NO forms ROONO first, which might decompose to RO + ONO or isomerize to RONO₂. Even though barrier for rearrangement of ROONO is lower than RO + NO₂, the scheme here seems to exaggerate the role of rearrangement. RONO₂s were missing in Fig S16 and Table S9.

Author comment: We agree that BPR + NO would form ROONO first, that might directly decompose to RO + ONO or isomerize to the more stable RONO₂. However, that is not the focus of our work. Our focus is on the eventual fate of the fraction of RO₂ + NO reactions that lead to the more stable RONO₂ organo nitrate. This is why the PES in Fig. S16 and Table S9 are missing the ROONO intermediate (we assume that the reviewer meant ROONO and not RONO₂ in their comment?). Additionally, we should have noticed (and mentioned) that the rearrangement rates in Table S9 are of *thermalized* RONO₂s as is probably obvious from the rates and the corresponding lifetimes of tens of minutes to hours. We acknowledge that showing the PES in Fig. S16 starting from reactants RO₂ + NO without clarifying the lack of role of excess energy in the rearrangement rates is misleading. If excess energy played a role in the rearrangement of RONO₂, then we agree with the reviewer that including the ROONO intermediate in the PES would have been necessary. Fortunately, the rearrangement is in fact thermal, and we can therefore avoid computing the complex PES involved in the isomerization of ROONO to RONO₂. We now state explicitly that 1) the RONO₂ rearrangement reaction is thermal and 2) that the general PES of RO₂ + NO includes the ROONO intermediate but is excluded in Fig. S16 and Table S9 for simplicity as we only consider the fate of the more stable RONO₂.

Changes to supplementary: The presence of the ROONO intermediate in RO₂ + NO reaction is mentioned and discussed in the supplementary text and in Fig. S16 caption.

“The previous mechanistic understanding of aromatic oxidation was that the primary fate of BPRs under polluted conditions is bimolecular reactions with NO to form B-ROONO intermediates that can either decompose into alkoxy radicals or isomerize to form organo nitrates (B-RONO₂; BPR + NO => B-RONO₂; B = bicyclic). While only some fraction of BPR + NO reactions will lead to B-RONO₂, these organo nitrates are thought to be stable once formed.”

“MESMER simulations indicate that even when the B-ROONO intermediates are neglected, excess energy from RO₂ + NO => B-RONO₂ step does not affect the B-RONO₂ dissociation rates: the dissociation reactions are fully thermal.”

“Similarly, the atmospheric lifetimes of Tol-meta-B-RONO₂ and Tol-para-B-RONO₂ are about 28 minutes and 3 hours, respectively.”

Fig. S16 caption: *“Note that BPR + NO will first lead to a ROONO intermediate and only a fraction will subsequently isomerize to the more stable RONO₂ isomer (the rest directly decomposing into RO + NO₂). We only include RONO₂ here as we are interested in the fate of the intermediate that is generally considered to be stable under atmospheric conditions.”*

4. Presumably, C₇H₉O₉ and C₇D₂H₇O₉ were from reaction of C₆H₅CH₃, and C₇D₃H₆O₉ and C₇D₅H₄O₉ were from reaction of C₆H₅CD₃ (Table S6). So the C₇H₉O₉ contains only two exchangeable H-atoms, instead “two, three and four” (Page S14). Please confirm or explain.

Author comments: It is correct that the dominant C₇H₉O₉ signal has two exchangeable H-atoms, and we indicate this in the original supplementary text as follows: “Similarly, the proposed autoxidation mechanism points to the C₇H₉O₉ peroxy radical isomer *with two acidic functional groups* that forms rapidly and is likely the dominant signal. This is corroborated by the D₂O experiment.” When we note the two, three and four H->D exchanges, it is for the C₇H₉O₁₁ peroxy radical, as noted in the original supplementary text. We have now modified the sentence to say 11-oxygen containing C₇H₉O₁₁ peroxy radical to make it clear that we are talking about the O₁₁ peroxy radical.

Changes to supplementary: In the supplementary, “*The D₂O experiment also indicates three C₇H₉O₁₁ peroxy radical isomers with two, three and four H->D exchanges.*” changed to “*The D₂O experiments also indicate three isomers for the 11-oxygen containing C₇H₉O₁₁ peroxy radical with two, three and four H->D exchanges.*”

5. In Fig 2, three C₉H₉O₉ isomers were given as R³-Epo-OOH-R’O₂ (containing one exchangeable H), R^{2b}-OOH-R’₂O₂ (containing three exchangeable Hs), and the green C₇H₉O₉ (lower left corner, containing two exchangeable Hs). But R^{2b}-OOH-R’₂O₂ has a fast isomerization rate of ~230 s⁻¹, and the last isomer would have a fast H shift (from the aldehyde H). Alternative isomers of C₇H₉O₉ might be needed to account for the observed signal.

Author comments: We agree that R^{2b}-OOH-R’₂O₂ is unlikely to accumulate due to its fast isomerization rates. This could at least partially explain why we do not see C₇H₉O₉ isomers with 3 exchangeable H-atoms (see Fig. S14 and S15 where the isomer with 2 exchangeable H atoms is clearly the dominant peak). We also note in the manuscript that we measure C₇H₉O₉ with two exchangeable H-atoms (this sentence is slightly modified to note that the *dominant peak corresponding to C₇H₉O₉* has two exchangeable atoms as some small contribution of a peroxy radical isomer with three exchangeable H-atoms cannot be ruled out). It is a good idea to explicitly acknowledge that the blue pathway is not a source of the C₇H₉O₉ peroxy radical exactly for the reason the reviewer suggests. We now note that in the manuscript.

Changes to manuscript: Section Experimental results, 1) “*This is more consistent with the mechanism we propose because the formed O₉-RO₂ via the dominant green pathway in Fig. 2...*” 2) “*The blue pathway in Fig. 2 does lead to a O₉-RO₂ with three exchangeable H-atoms, but likely due to its fast isomerization reaction (~230 s⁻¹) rapidly forms O₁₁-RO₂, and is not measured.*”

6. Page S14, “... Fig S11 shows the time series of the normalized signals ...”: The figure cannot be found.

Author comments: Thank you for pointing this out. We initially included and later deleted the time series plot for the D₂O experiments, but then neglected to remove our reference to it in the supplementary text. We decided that the figure was too complicated while adding little value. The figure is below:

We will remove the reference to the missing figure in the supplementary text. Alternatively, if the reviewer suggests that the plot should be included, we will add it to the supplementary text.

Changes to supplementary: Removed the reference to Fig. S11 in the supplementary text.

7. Kinetics of R1 (as in Fig S17 and S18): It should be noticed that the two additions of O₂ are both reversible, particularly for R1a-RO₂ and when R1 contains excess energy. The usual "+ O₂ – HO₂" process in alpha-hydroxyl alkyl radicals may be limited by the 1,4 H-shift because the addition of O₂ to R1a-RO₂ is much less exothermic (only ~7 kcal/mol here) than the addition to normal alkyl radicals such as CH₃CHOH radical (exothermic by ~30 kcal/mol).

Author comments: We agree that the reversibility of the O₂ addition reactions should be noted. The lower exothermicity of the formation of R_{1a}-RO₂ will also likely favor the formation of R_{1b}-RO₂, making the green channel in Fig. 2 more important. This is now noted in the supplementary text. In addition, we included R_{1a}-RO₂ + NO => R_{1a}-ROONO and R_{1b}-RO₂ + NO => R_{1b}-ROONO sink reactions with rate coefficients of 1 s⁻¹ to include the reverse reactions of R_{1a}-RO₂ and R_{1b}-RO₂ in the MESMER simulation. The ROONO was a model system generated without a conformer sampling step and computed at the B3LYP/6-31+G(d) level. The exothermicity of the formation of ROONO was arbitrarily set to -15 kcal/mol relative to the RO₂s. If no sink reactions are added and R_{1a}-RO₂ and R_{1b}-RO₂ are simply treated as "modelled" to account for the reverse reactions, then the product that is thermodynamically the most stable is formed with a 100% yield. Therefore, a sink with an approximately correct timescale to correctly model the effect of reversibility is needed. A more accurate description of the ROONO sink was considered unnecessary as long as the timescale is correct. The rate of conversion of RO₂ to ROONO was set at 1 s⁻¹ by giving a NO concentration of 5×10¹⁰ molecules/cm³ and a reaction rate coefficient of 2×10⁻¹¹ cm³/molecule/s. This simulation produced a 100% yield of R_{1b}-ROONO, indicating a larger

yield of $R_{1b}\text{-RO}_2$ when the low exothermicity of $R_{1a}\text{-RO}_2$ formation and consequently the reversibility of the two O_2 addition reactions are accounted for.

Changes to supplementary: *“The formation of all intermediates in Fig. S18 are preceded by transition states and these reactions were treated using the SimpleRRKM method with Eckart tunneling. The intermediates $P\text{-C}_2$ and R_1 were treated as “modelled” and $R_{1a}\text{-RO}_2$ and $R_{1b}\text{-RO}_2$ as sinks in the simulation. The MESMER input file is provided in the data archive. This resulted in a 20% yield for $R_{1b}\text{-RO}_2$. Note that the F12 correction causes TSi to be slightly negative, and while this does not impede the MESMER simulations, it does add some uncertainty to the predicted branching ratios. Also, the two O_2 addition reactions are reversible, which is especially important for $R_{1a}\text{-RO}_2$ given that its formation is less exothermic and R_1 contains excess energy. We accounted for the reverse reactions of $R_{1a}\text{-RO}_2$ and $R_{1b}\text{-RO}_2$ by including a $\text{RO}_2 + \text{NO} \Rightarrow \text{ROONO}$ channel with a loss rate of 1 s^{-1} . This was done by adding a model ROONO sink to $R_{1a}\text{-RO}_2$ and $R_{1b}\text{-RO}_2$ using the SimpleBimolecularSink method in MESMER with NO concentration of $5 \times 10^{10} \text{ molecules/cm}^3$ and a reaction rate coefficient of $2 \times 10^{-11} \text{ cm}^3/\text{molecule/s}$. $R_{1a}\text{-RO}_2$ and $R_{1b}\text{-RO}_2$ were treated as “modelled” to account for their reverse reactions. If no sink reactions are added and $R_{1a}\text{-RO}_2$ and $R_{1b}\text{-RO}_2$ are simply treated as “modelled” to account for reverse reactions, then the product that is thermodynamically the most stable is formed with a 100% yield. Therefore, a sink with an approximately correct timescale to correctly model the effect of reversibility is needed. The precise details of the sink do not matter as long as the timescale is correct. Therefore, a model ROONO system computed at the B3LYP/6-31+G(d) level was used with an effective rate of 1 s^{-1} . The MESMER input PES is shown in Fig. S19 and the input file is provided in the data archive. This resulted in 100% yield of $R_{1b}\text{-ROONO}$, indicating a larger yield of $R_{1b}\text{-RO}_2$ when low exothermicity of $R_{1a}\text{-RO}_2$ formation is accounted for.”*

8. Flow tube simulations: Could typical concentrations of OH radical and RO_2 be given here? These quantities are essential in assessing the competition between unimolecular and bimolecular reactions for peroxy radicals.

Author comments: These are now included in the supplementary. The OH concentrations are $\sim 7 \times 10^7 \text{ molecules cm}^{-3}$ and RO_2 concentrations are $\sim 9 \times 10^9 - \sim 1 \times 10^{10} \text{ molecules cm}^{-3}$ for the flow tube simulations with reaction times of 0.8 s, 1.5 s and 3.7 s.

Changes to supplementary: Typical concentrations of OH radical and RO_2 now provided in Fig. S20 in the supplementary.

9. Minor comments: # Page S9, the 4th line from Bottom: “Fig S4” should be “Fig S9”; Page S10, “Fig S5” should be “Fig S10”; Page S11: “Fig S6” should be “S11”; ...

Author comments: Thank you for pointing these out. They have now been corrected.

Changes to supplementary: Corrections have been made to the supplementary.

Reviewer 3:

General comment: This manuscript describes a pathway in the atmospheric oxidation of aromatics that appears to be the missing link between the traditional chemistry and the unexplained HOM formation from these compounds. By combining theoretical kinetic calculations, a kinetic model, and experimental observations, the authors make a case that inclusion of this pathway leads to predictions that are consistent with the observations. The research is timely, as there is a lot of interest in HOM and aerosol formation, and the role of aromatics in urban air quality. The methodologies used are of high level and appropriate for the topic, leading to robust results. This paper describes a high-impact result that has the potential to solve a long-standing conundrum. I support publication of this manuscript.

Author comment: We thank the reviewer for their positive comment and their support for the publication of the manuscript.

First main comment: To me, it feels that the authors did not do due diligence to the literature. While I recognize that space and number of references are limited in the main text, the supporting information contains extensive descriptions of mechanistic steps without any referencing at all. Most of this even seems to be based on a single paper (Wang and coworkers), which does not acknowledge the vast effort that the community has put into elucidating aromatics oxidation mechanisms to establish a base upon which the current study can add its contribution. The paper also generalizes to aromatics beyond toluene, which is certainly not covered by a single reference to Wang et al. Reviewing the aromatics literature is daunting and I understand that this is not the manuscript to do this in full, but at the very least some key papers should be cited, as well as mechanistic reviews (e.g. Vereecken 2019, Vereecken and Francisco 2012, Atkinson 2003, Calvert 2002,...). These, and perhaps some additional references to SARs, will also support some of the calculations in the SI that were only done using low-level methodologies. Below, I will not list all places where literature citations are needed, as it is pervasive.

Author comments: We acknowledge that the aromatic oxidation mechanisms in the supplementary should have been better cited. Key papers suggested by the reviewer are now cited where appropriate.

Changes to supplementary: Papers are cited in supplementary. Page S4: *"The initial steps of OH addition to toluene is shown in Fig. S4 and have been discussed in detail in previous works (16-19)."*

Second main comment: One should be weary about presenting the experiments as proof of the contribution of the characterized pathway. The experiments show that an autoxidation chemistry exists, but does not establish that the molecules observed are those predicted by the theoretical analysis; autoxidation is known to generate many isomers. While the results seem consistent, there remain uncertainties on the theoretical predictions and the experimental observations, and the latter only span a small range of reaction conditions that may be insufficient to discriminate between distinct reaction mechanisms. Likewise, the authors do not establish that the proposed mechanism is also consistent with the observations.

available in the literature.

Related to this is that, at first glance, the mechanism seems to miss some pathways, or at least does not discuss their contribution as far as I could find. This includes e.g. the reversibility of the epoxidation in R3 (competitive against O2 addition), the H-migration from -OOH to -O-radical in R3 (with a literature rate $\sim 1E10$ s⁻¹), or H-scrambling and ring closure in R2b-OOH-R'2O2, all of which represent reactions classes known to be fast. The kinetic calculations also do not seem to account for the impact of fast H-scrambling across OOH/OO groups on the rate coefficient, as well-documented in the literature. It is unclear whether any of this would affect the predictions, but perhaps the mechanism is not fully robust yet.

Author comment: We agree that we must be careful when assigning peaks from experimental spectra to one theoretically predicted mechanism and we acknowledge that the proposed mechanism does not represent the entire autoxidation chemistry of toluene. It is in fact impossible for it to do so given that we have not explored all the autoxidation channels even in the mechanism we propose (i.e., the autoxidation of the P-C₁ peroxy radical). Our choice to discuss the experimental results in relation to our mechanism were to 1) offer mechanistic insights on the autoxidation products of toluene + OH we measure that are formed under the rapid reaction time conditions of our experiments, and 2) highlight that multiple autoxidation pathways can occur concurrently, as predicted by our mechanism. Regarding 1) the only experimental observation available in literature carried out under the reaction time conditions that are close to our study is Wang et al. 2017, and we have compared our result to theirs extensively. Our D₂O experiments that indicate the number of -OH and -OOH functional groups are not in disagreement with the RO₂ isomers from our mechanism, and this adds credence to our mechanism. Regarding 2) This result is non-trivial as no previous work has shown that excess energy from the formation of an RO₂ leads to multiple autoxidation pathways. Specifically, the two isomers of O₇-RO₂, one with one labile H-atom and another with two labile H-atoms shows that multiple autoxidation paths occur concurrently, and this is not in disagreement with the green and red pathways in Fig. 2.

We acknowledge that the proposed pathways provide *one* explanation and possible structures of the observed highly oxidized RO₂s. We agree that there could be other autoxidation pathways not considered in this study that can also lead to the RO₂s we detect. This clarification has now been made in the text.

Regarding the red channel in Fig. 2 and the issues the reviewer raises about 1) reversibility of R₃-Epo-R' and 2) H-migration from -OOH to -O radical, which is essentially the reverse reaction of R₃ back to P-C₂. We ran a MESMER simulation along the R₁, R₂ and R₃ pathways of P-C₂ with parameters calculated at the ωB97X-D/aug-cc-pVTZ level. The lower level of theory is to use the same methodology as what is used for computing the TS to R₃ for which CCSD(T)-F12 calculations are unreliable. For R₁, we ignored the closed-shell C₇H₈O₅ forming channel and assigned the formation of R_{1b}-RO₂ as the direct sink of R₁ (+ O₂). The simulation shows that R₃-Epo-RO₂ is a significant product (36% yield). While the reverse reactions have low barriers, the excess energy of P-C₂ likely drives the formation of the R₃-Epo-R'O₂.

Other pathways available to $R_{2b}\text{-OOH-R}'_2\text{O}_2$ such as H-scrambling could be competitive with the vinoxy forming channel shown in Fig. 2, but these do not directly lead to the next O_2 addition. The -OH H-atom is likely still very labile, and abstraction of that H-atom by the H-scrambled RO_2 would still lead to the same $R_{2b}\text{-OOH-vinoxy}$. The ring closure reactions are unlikely to be competitive with H-shifts from -OH and -OOH groups as our previous calculations, e.g., ring closure reactions of $R_{2a}\text{-RO}_2$, indicate that these have much larger barriers. We completely agree with the reviewer that the mechanism is not robust yet. We now make that clear in the manuscript text.

Changes to manuscript: Pg. 5: " *R_3 and $R_3\text{-Epo-R}'$ both have small reverse barriers, but the yield of $R_3\text{-Epo-RO}_2$ is nevertheless likely to be high due to the excess energy of $P\text{-C}_2$ as shown in our simulations (see supplementary section S12).*"

Pg. 5. "*Note that autoxidation is known to produce many isomers and the pathways shown in Fig. 2 are not the only autoxidation pathways occurring in our measurements. However, these are the only pathways that can currently explain the observed H-to-D shifts in the isomers we measure.*"

Changes to supplementary: New supplementary section S12 added that describes the favorability of the formation of $R_3\text{-Epo-RO}_2$.

Minor comments:

1. p. 3 line 79: "finite timescales"
define "finite", in this context probably relative to atmospheric transport time scales (regional, continental, hemisphere, global ?)

Author comment: Thank you, we agree that the word "finite" is somewhat ambiguous in this context. We have now changed it to *regional atmospheric transport timescales*.

Changes to manuscript: Pg. 2: "*Intriguingly, several of the organo nitrates and other closed shell products derived from the *i*-BPR are likewise unstable and decompose on regional atmospheric transport timescales.*"

2. p. 4 lines 90-105:
This would be significantly easier to understand with a graphical inset with a Lewis-structure representation of an example.

Author comment: We have now included a 2-D schematic of the rearrangement reaction in Fig. 1.

Changes to manuscript: A 2-D schematic of the rearrangement reaction is included as Fig. 1 A.

3. p. 5 fig 1:
Top: One can not really see what the rearrangement is, due to the use of ball-and-stick graphics. See also remark above.
Bottom: Perhaps an Arrhenius plot with x-axis in $1000\text{K}/T$? Then again, Nat.Comm is not a kinetics journal.

Author comment: Regarding top: we have now included a 2-D schematic of the rearrangement reaction in Fig. 1 A. Regarding bottom: We decided to show an easier to read temperature relation plot for modelers. The Arrhenius form of the plot is now included in the supplementary (Fig. S22).

Changes to supplementary: The Arrhenius plot of the rearrangement reactions of toluene derived i-BPR is provided in the supplementary (Fig. S22).

4. p 7 fig 2:

- Caption: "autoxdation" -> autoxidation
- The intermediates must not be represented as trans-alkenes but strictly as cis-isomers or unspecified (where allylic rotation is possible). Trans-stereoisomers are not what was calculated (sampling a few of the log files shows as much), are not what is expected for the parent molecule decomposition, and would prevent some of the chemistry to happen (e.g. the H-shift in R2b-RO2)
- The fates for P-C2 do not sum to 100%

Author comment:

- Typo in caption corrected.
- The intermediates in Fig. 2 are now represented as cis-isomers.
- The fates of P-C₂ should sum up to 79% as the remaining is the yield of P-C₁. We acknowledge that this might not be clear from the figure, and we now make a note to this effect in the figure caption.

Changes to manuscript:

- Typo corrected.
- Fig. 2 modified to show intermediates as cis-isomers.
- Text added to Fig. 2 caption noting that the yield of P-C₂ is 79% and that the fates of P-C₂ add up to that.

5. p. 9 line 183: " indicating that the mechanism proposed in this work is the dominant unimolecular pathway to the SOA precursors that we detect."
"is" -> is likely to be / is consistent with / ...

Author comment: We agree with the reviewer's correction to this sentence. The text now reads "...indicating that the mechanism proposed in this work *is likely to be* the dominant unimolecular pathway to the SOA precursors that we detect."

Changes to manuscript: The text now reads "...indicating that the mechanism proposed in this work *is likely to be* the dominant unimolecular pathway to the SOA precursors that we detect."

6. p. 12, fig 4 : perhaps remove the minor grid lines to lighten the plot

Author comment: The minor grid lines have been removed from Fig. 4.

Changes to manuscript: Minor grid lines in Fig. 4 have been removed.

7. p. 13, line 246: "This could re-initiate autoxidation and provide a long range transport mechanism for NO_x"
It is unclear how prompt decomposition losing NO₂ could be useful for long-range NO_x transport, as the NO₂ is then lost before it can be transported?

Author comment: While the decomposition of the ipso-RONO₂ is prompt, the lifetimes RONO₂s derived from the BPR of benzene and meta and para isomers of toluene are on the orders of tens of minutes to hours as indicated by their decomposition rates in the last row of Table S9. However, we agree that these timescales are too short for *long-range* transport, so we have decided to omit "long-range" and only say "transport" instead.

Changes to supplementary: "The relatively fast conversion of the benzene derived B-RONO₂ to RB-RO + NO₂ is significant as it corresponds to an atmospheric lifetime of Ben-B-RONO₂ of about 34 minutes. *Similarly, the atmospheric lifetimes of Tol-meta-B-RONO₂ and Tol-para-B-RONO₂ are about 28 minutes and 3 hours, respectively.*"

Changes to manuscript: "long-range" deleted in two instances, pages 2 (introduction) and 7 (conclusion).

8. p. 16, wave function stability
What do the CASSCF calculations show for multi-reference effects on the biradical O-O bond breaking? Is the active space (1 virtual orbital) large enough to accommodate such multi-reference effects ?

Author comment: The wave function stability calculations were only carried to ensure that the initial Hartree-Fock calculation before CCSD(T) converged to the correct minima and not to check for multi-reference effects. But we acknowledge that we should have noted the possibility of multi-reference effects in the rearrangement mechanism. We now provide the T1 diagnostic numbers of the T-C₁ and T-C₂ rearrangement transition states in the supplementary. Note that these are within 0.025 and under the accepted threshold for open-shell systems. In addition, the triples contribution to the total atomization energy %TAE(T) value was calculated for T-C₁ and found to be 2.7%, indicating minor static correlation effects.

Changes to supplementary: The T1 diagnostic and %TAE(T) numbers are now provided in Section 11 of the supplementary.

9. p. 17, line 341 "In the MESMER simulations, the harmonic frequencies were treated as hindered rotors"
I assume the authors mean that only those internal modes corresponding to internal rotations were removed from the vibrational frequencies and treated as hindered rotors.

Author response: The reviewer is correct. The modified text now reflects the reviewer's description.

Changes to manuscript: *"In the MESMER simulations, the internal modes corresponding to internal rotations were removed from the vibrational frequencies and treated as hindered rotors. This was performed..."*

10. p S9, top, ring opening in B-alkyl

There are much better discussions of this channel available that should be cited, e.g. Glowacki et al. 2009, Vereecken 2018, Vereecken 2019, as well as experimental evidence of its lack of importance by the Wennberg group (Xu et al. 2020)

Author response: Thank you. The suggested papers are now cited.

Changes to supplementary: The suggested papers on the ring opening of B-alkyl are now cited in the supplementary. *"In addition to adding an O₂ to form BPR, B-alkyl can undergo an isomerization reaction to form B-oxy (see Fig. S8 (2-4)). This path has been shown to be unimportant for the benzene derived BPR(6)."*

11. p. S12, NO experiments.

NO can lead to autoxidation proceeding through different pathways (so-called alkoxy-peroxy autoxidation steps). They may form different isomers of the same mass, and a mass spectrum won't reveal mechanistic changes. Alkoxy-peroxy autoxidation is also implicated as a key step in ring breaking HOM formation in some VOC (e.g. Shen et al. 2022, Guo et al. 2022). Furthermore, Vereecken (2018, 2019) points out the role of NO in changing the early stage aromatic oxidation mechanism, based on available literature data. These experiments may thus not as straightforward to interpret as the authors imply.

Author comment: We agree with the reviewer that the formation of organo nitrates is not the only possible pathway for RO₂ + NO. Alkoxy-peroxy autoxidation steps are indeed important in many cases but are expected to lead to products with one less O-atom. In our case, this would lead to peroxy radicals with even number of oxygen atoms, not odd as measured. It is unlikely that the peroxy radicals will react with NO two times to produce odd number of O-atoms and react with NO a third time to produce the organo nitrates we detect under the 3.7 s reaction time conditions of our experiments.

Changes to manuscript: *"Reactions of peroxy radicals with NO can also lead to alkoxy radicals and the molecule can continue to oxidize via the alkoxy-peroxy autoxidation pathway. However, the resulting peroxy radicals will have an even number of oxygen atoms and not odd as the peroxy radicals involved in the formation of the organo nitrates reported in this study. The peroxy radicals are unlikely to undergo two NO reactions to revert to an odd number of oxygen atoms followed by a third NO reaction to produce the organo nitrates we detect under the short residence time conditions of our experiments."*

12. p. S17, top: "the rearrangement mechanism is also possible for other aromatic derived bicyclic molecules. Table S8 "

Without context, it is not clear which rearrangement mechanism is meant. The bicyclic alkyl radicals have very different rearrangement mechanism reported (ring

retaining with epoxide-alkoxy) than given in this work for bicyclic peroxy radicals (ring opening with H-shift and carbonyl formation).

Author comment: We agree that additional clarification is needed here. We now explicitly mention that the rearrangement mechanism we discuss is the one reported in this work.

Changes to supplementary: The text in the supplementary is modified to show that the rearrangement mechanism described is the one reported in this work. *"In addition to BPRs, the rearrangement mechanism described in this work is also possible for other aromatic derived bicyclic molecules."*

13. p. S19, bottom

- Mention "site-specific O₂ addition" at the start of the last paragraph
- "H-abstraction by O₂ to form C₇H₈O₅ is not direct but follows the initial addition of O₂ to the C(OH) carbon."

This needs specific literature citations

- "These transition states were calculated to be -0.7 kcal/mol and 1.9 kcal/mol, respectively, above R₁ at F12 level of theory." -> "...relative to R₁ at the F12 level of theory."

- "the potential energy surface shown in Fig. S12 was used as input in a MESMER simulation to account for the excess energy of R₁."

This glosses over a lot of details. At the very least refer explicitly to the Mesmer input files in the Zenodo archive, but preferable document this more thoroughly.

Author response:

- Thank you, we now mention this.
- This was a result from our calculations. Intrinsic reaction coordinate calculations on the transition state structure corresponding to the H-abstraction by O₂ showed that the reactant well structure has O₂ on the C(OH) carbon. We now make this clear in the supplementary text.
- We have incorporated this correction.
- Additional details are added and the MESMER input is added to the Zenodo archive.

Changes to supplementary: Pg. S19: *"Due to site specific O₂ addition..."*

"Transition state calculations followed by intrinsic reaction coordinate calculations indicated that the H-abstraction by O₂ to form C₇H₈O₅ is not direct but follows the initial addition of O₂ to the C(OH) carbon."

"The formation of all intermediates in Fig. S18 are preceded by transition states and these reactions were treated using the SimpleRRKM method with Eckart tunneling. The intermediates P-C₂ and R₁ were treated as "modelled" and R_{1a}-RO₂ and R_{1b}-RO₂ as sinks in the simulation. The MESMER input file is provided in the data archive."

14. p. S20, bottom "the analytical solution is represented by its Taylor expansion"
The analytical solution to what? The MCM ?

Author response: The analytical solution for diffusion of chemical species in a laminar flow field of a tube, as it is provided by Ingham (D. B. Ingham, *Journal of Aerosol Science* 6, 125-132 (1975)), is not a good representation if there are sources/sinks of those molecules (we got sources: the chemical reactions). As a consequence, we expanded this analytical solution into its Taylor-representation (the same is done in the referenced article: Pichelstorfer & Hofmann, 2017 - Pichelstorfer, L. and Hofmann, W., *Aerosol Science and Technology* 51, 1419-1428 (2017)).

Changes to supplementary: *"...the analytical solution for diffusion of chemical species in a laminar flow field of the tube is represented by its Taylor expansion"*.

15. p. S21, middle "The likely reason for this is the less reactive TME RO2s."
Why are they less reactive ? Is literature data available ?

Author response: The sentence was added in error, it has now been removed.

Changes to supplementary: Sentence "The likely reason for this is the less reactive TME RO2s" removed in the supplementary.

16. p. S22, ref 1: remove "Vertiotie.4P"

Author response: Thank you for pointing out the typo. This has now been removed.

Changes to supplementary: Typo removed from supplementary.

Reviewers' Comments:

Reviewer #2:

Remarks to the Author:

Comments on NCOMMS-23-12910A, "Molecular rearrangement of bicyclic peroxy radicals: key route to aerosol from aromatics", by Iyer et al.

The revision here provides more evidence on the proposed new mechanism. The oxidation mechanisms are always complex, and search for new reaction pathways is desired all the time. The finding here provides additional information on the formation of highly oxidized products and SOAs.

Comments:

1. It should be noticed that the rearrangement of BPRs from ipso-addition in this study and the H shift of BPRs from para-addition from Wang et al. are parallel and independent reaction pathways. Under different conditions (particularly the residual time), pathways with different rates show different product formation. These two pathways do not compete with each other. Under the atmospheric conditions, there is enough time for the reaction to complete and both pathways should exist and different products are formed. When NO is low, both pathways are likely the dominant for ipso- and para-additions.

We agree the existence of the new reaction pathway, but the H shift mechanism in para-path should be at least equally important in real atmosphere.

2. On BPRs with NO: The existence of ROONO does not affect the kinetics of RONO₂. I am satisfied with the statement "... only some fraction of BPR + NO reactions will lead to B-RONO₂". Decomposition of x-B-RONO₂ is fully thermal, so the RRKM-ME modeling does not need the entry "x-BPR + NO" as shown in S16.

Reviewer #3:

Remarks to the Author:

This manuscript describes a pathway in the atmospheric oxidation of aromatics that, based on the presented theoretical and experimental data, is likely to be the source of the unexplained HOM formation from such compounds. The theoretical methodologies used are expected to yield robust results, and are supported by a thoughtful interpretation of experimental data.

The research is important and timely, as there is a lot of interest in HOM and aerosol formation, and the role of aromatics in urban air quality. This pathway is likely to receive a lot of further attention, and will be included in atmospheric kinetic and air quality models.

The authors have addressed all questions and comments raised in the referees comments, and I have no further suggestions for improvement. I support publication of this manuscript as is.

We thank the reviewers once again for their time and feedback.

Reviewer 2:

General comment: The revision here provides more evidence on the proposed new mechanism. The oxidation mechanisms are always complex, and search for new reaction pathways is desired all the time. The finding here provides additional information on the formation of highly oxidized products and SOAs.

Author comment: We thank the reviewer for their positive comment.

1. It should be noticed that the rearrangement of BPRs from ipso-addition in this study and the H shift of BPRs from para-addition from Wang et al. are parallel and independent reaction pathways. Under different conditions (particularly the residual time), pathways with different rates show different product formation. These two pathways do not compete with each other. Under the atmospheric conditions, there is enough time for the reaction to complete and both pathways should exist and different products are formed. When NO is low, both pathways are likely the dominant for ipso- and para-additions.

We agree the existence of the new reaction pathway, but the H shift mechanism in para-path should be at least equally important in real atmosphere.

Author comment: We agree that the pathway described in the Wang et al. 2017 paper, despite not being observed in our experiments, could nevertheless be important under certain atmospheric conditions. We now note this in the manuscript.

Changes to manuscript: Page 3 - *Nevertheless, if this is the dominant fate of the toluene + OH system in our experiments,...*

Page 3 - *However, we observe that the mass spectra for toluene and CD₃-toluene are near identical, indicating that H-shift from the methyl group is unlikely to play a major role in our experiments. Nevertheless, this pathway might well be important under certain atmospheric conditions. The methyl H-abstraction pathway reported in Wang et al. and the rearrangement pathway reported here do not compete with each other but depend on the site of the initial OH addition to toluene (para vs ipso relative to the methyl substituent).*

2. On BPRs with NO: The existence if ROONO does not affect the kinetics of RONO₂. I am satisfied with the statement "... only some fraction of BPR + NO reactions will lead to B-RONO₂". Decomposition of x-B-RONO₂ is fully thermal, so the RRKM-ME modeling does not need the entry "x-BPR + NO" as shown in S16.

Author comment: We agree that Supplementary Figure 16 does not need the reactants x-BPR + NO. This has now been corrected.

Changes to supplementary: Supplementary Figure 16 modified to start the PES from x-RONO₂ instead of x-BPR + NO. Supplementary Table 9 has also been similarly modified.

Reviewer 3:

General comment: This manuscript describes a pathway in the atmospheric oxidation of aromatics that, based on the presented theoretical and experimental data, is likely to be the

source of the unexplained HOM formation from such compounds. The theoretical methodologies used are expected to yield robust results, and are supported by a thoughtful interpretation of experimental data.

The research is important and timely, as there is a lot of interest in HOM and aerosol formation, and the role of aromatics in urban air quality. This pathway is likely to receive a lot of further attention, and will be included in atmospheric kinetic and air quality models.

The authors have addressed all questions and comments raised in the referees comments, and I have no further suggestions for improvement. I support publication of this manuscript as is.

Author comment: We thank the reviewer for their positive comment.